# An alternative pathway for membrane protein biogenesis at the endoplasmic reticulum

Sarah O'Keefe [1✉], Guanghui Zong [2], Kwabena B. Duah[3], Lauren E. Andrews[3], Wei Q. Shi [3] & Stephen High [1✉]

The heterotrimeric Sec61 complex is a major site for the biogenesis of transmembrane proteins (TMPs), accepting nascent TMP precursors that are targeted to the endoplasmic reticulum (ER) by the signal recognition particle (SRP). Unlike most single-spanning membrane proteins, the integration of type III TMPs is completely resistant to small molecule inhibitors of the Sec61 translocon. Using siRNA-mediated depletion of specific ER components, in combination with the potent Sec61 inhibitor ipomoeassin F (Ipom-F), we show that type III TMPs utilise a distinct pathway for membrane integration at the ER. Hence, following SRP-mediated delivery to the ER, type III TMPs can uniquely access the membrane insertase activity of the ER membrane complex (EMC) via a mechanism that is facilitated by the Sec61 translocon. This alternative EMC-mediated insertion pathway allows type III TMPs to bypass the Ipom-F-mediated blockade of membrane integration that is seen with obligate Sec61 clients.

[1] School of Biological Sciences, Faculty of Biology, Medicine and Health, University of Manchester, Manchester, UK. [2] Department of Chemistry and Biochemistry, University of Maryland, College Park, MD, USA. [3] Department of Chemistry, Ball State University, Muncie, IN, USA. ✉email: sarah.okeefe@manchester.ac.uk; stephen.high@manchester.ac.uk

pomoeassin F (Ipom-F)[1] and mycolactone[2,3] are natural pro-
ducts that inhibit the Sec61-mediated translocation of newly
synthesised polypeptides into and across the endoplasmic reti-
culum (ER) membrane. Both compounds are substrate selective;
effectively blocking the translocation of secretory proteins and the
ER integration of type I and type II transmembrane proteins
(TMPs) (Fig. 1a), but not affecting two type III TMPs (glycophorin
C; GypC and synaptotagmin 1; Syt1) studied to date[1–3]. Whilst this
behaviour suggests that type III TMPs may not follow a typical
pathway for ER integration[1], earlier reconstitution studies suggested
that GypC could be membrane inserted via the Sec61 complex[4].
Notably, the membrane insertion of tail-anchor (TA) proteins
(Fig. 1a) at the ER is also unaffected by Ipom-F and mycolactone,
consistent with substantial evidence that they can exploit one
or more Sec61-independent pathways[5,6]. Given the effectiveness of
the in vitro blockade of other Sec61-dependent clients observed
with Ipom-F and mycolactone[1–3,] we therefore considered the
possibility that, like TA proteins, type III TMPs may also be able to
insert into the ER membrane via one or more Sec61-independent,
pathway(s).

The ER membrane complex (EMC) is evolutionarily conserved
and its disruption has wide-ranging and pleiotropic effects on
membrane protein biogenesis[7–15]. Furthermore, the EMC has
recently been shown to act as an ER membrane insertase[14,16,17]
that is alone capable of mediating the post-translational insertion
of certain TA proteins[18] and the co-translational insertion of the
first transmembrane domain of a multi-pass integral membrane
protein[14,19]. In the latter case when the first transmembrane
domain is studied in isolation it orients with its N-terminus on
the exoplasmic side of the membrane ($N_{exo}$), and on that basis it
might be viewed as a synthetic type III TMP (Fig. 1). However, in
the case of such multi-pass proteins, the insertion of subsequent
transmembrane domains is dependent on the Sec61 complex[19].
Hence, it appears that the EMC and Sec61 complex act together
to provide a co-ordinated site for the membrane integration of a
subset of multi-pass TMPs at the ER[14,17,19–21].

Given that ER-targeting and preliminary engagement of
ribosome-nascent chain complexes with the Sec61 translocon are
maintained in the presence of mycolactone[22], and that type III
transmembrane domains have been shown to be in close proxi-
mity to the Sec61 complex during their membrane insertion[23], a
'sliding' model[24] has been proposed to explain the co-ordinated
actions of the Sec61 complex and the EMC. In this scenario,
following ribosome-nascent chain docking to the Sec61 complex,
the first transmembrane domain of some multi-pass TMPs may
insert headfirst ($N_{exo}$) into the ER membrane by using the EMC at
a location that is close to, but distinct from, the Sec61 lateral gate
that normally mediates transmembrane domain insertion[14,19,24].
In the absence of the EMC, it is proposed that a small proportion
of such headfirst transmembrane domains, particularly those with
a more hydrophobic character, may be able to access the Sec61
lateral gate directly, although this interaction may result in an
inverted topology[19,24].

Here, we explore the role of the Sec61 complex and the EMC
during membrane insertion by using a small panel of naturally
occurring single-pass type III TMPs, comprising one viral and
four mammalian proteins. We find that all five of our model type
III TMPs are efficiently inserted into ER-derived microsomes
even in the presence of Ipom-F[1]. Using semi-permeabilised (SP)
mammalian cells depleted of Sec61, EMC and/or signal recogni-
tion particle-(SRP) receptor subunits as an alternative source of
ER membranes[25,26], we then probed their respective contribu-
tions to type III TMP biogenesis. Our data suggest that the ER
insertion of type III TMPs typically requires the combined actions
of both the Sec61 complex and the EMC, which act downstream
of an SRP-dependent delivery step.

## Results

**Type III TMPs are resistant to Ipom-F**. To explore the biogenesis
of type III TMPs (Fig. 1a), we first used a well-established in vitro
system supplemented with ER-derived canine pancreatic micro-
somes (Fig. 1b) and exploited the Sec61 translocation inhibitor
Ipom-F (Fig. 1c)[1]. Using ER lumenal N-glycosylation as a reporter
for authentic membrane translocation[1,2], we analysed the mem-
brane integration of a small, yet biochemically diverse, panel of
model type III TMPs in the presence and absence of Ipom-F. These
model type III TMPs are: human immunodeficiency virus protein
Vpu (HIV-Vpu), small cell adhesion glycoprotein (SMAGP), gly-
cophorin C (GypC), tumour necrosis factor receptor superfamily
member 17 (BCMA) and synaptotagmin 1 (Syt1), in most cases
modified to include N-glycosylation sites (see Fig. 1d and accom-
panying legend).

Following the resolution of radiolabelled products by SDS-
PAGE, we found that all five-model type III TMPs displayed
apparently normal levels of N-glycosylation in the presence of 1
μM Ipom-F (Fig. 1d, lane 1 versus lane 3 in each panel). Likewise,
the integration of two C-terminally tagged TA proteins was
unaffected by Ipom-F (Supplementary Fig. 1a), consistent with
their Sec61-independent biogenesis[5]. In contrast, 1 μM Ipom-F
substantially inhibits the ER translocation of secretory proteins
and the integration of type I and type II TMPs (Supplementary
Fig. 1b–d)[1]. Hence, Ipom-F consistently inhibits Sec61-mediated
translocation in a substrate selective manner that emulates the
actions of mycolactone[1,2,27] and suggests that the integration of
type III TMPs is mechanistically distinct from other classes of
single-pass membrane proteins.

We next prepared SP cultured cells of human origin[25,28] and
compared the effects of Ipom-F to those in canine pancreatic
microsomes. Excluding qualitative changes in the efficiency of
substrate N-glycosylation, the effects of Ipom-F were directly
comparable between microsomes and SP cells across the range of
substrates analysed (Supplementary Fig. 1E). Hence, type I and type
II TMPs were Ipom-F sensitive whilst a type III TMP was not. We
conclude the effects of Ipom-F on Sec61-mediated protein
translocation (Fig. 1 and Supplementary Fig. 1)[1] is unaffected by
the source of ER-derived membrane[29,30].

**The effects of Sec61α and EMC5 knock-down on membrane
insertion are substrate selective**. We next probed the roles of the
Sec61 complex and the EMC (Fig. 2a) in the membrane integration
of type III TMPs using SP cells transiently depleted of each complex.
Using well-characterised siRNAs, we targeted the central component
of the Sec61 complex[31,32], Sec61α (Fig. 2b, c, lane 2)[26], and two
subunits of the EMC, the cytosolic EMC2 (Fig. 2b, c, lane 3)[33] and
the membrane-embedded EMC5 (Fig. 2b, c, lane 4)[18], for knock-
down and determined the effects on protein levels (Fig. 2c) using
quantitative immunoblotting (Fig. 2d). Multimeric human EMC has
seven membrane-embedded and two cytosolic subunits (Supple-
mentary Fig. 2a) and siRNA-mediated knock-downs of EMC2 and
EMC5 destabilise the wider EMC[10–14,18–20]. Hence, we also immu-
noblotted for the EMC6 subunit, together with the OST48 subunit of
the oligosaccharyl-transferase (OST) complex, which served as an ER
localised control[34], in order to confirm the specificity of our target
knock-downs.

Each knock-down selectively reduces expression of the target
without wide-ranging effects on ER membrane protein stability
per se (Fig. 2c, d) and has little (Sec61α knock-down) or
no (EMC2 or EMC5 knock-down) impact on OST activity[10,26],
as judged by levels of OST48, a type I TMP and presumed
Sec61 client (Fig. 2b–d). Furthermore, based on our analysis of
EMC6 levels, we conclude that knock-down of EMC2, and, to
an even greater extent, EMC5 is sufficient to destabilise

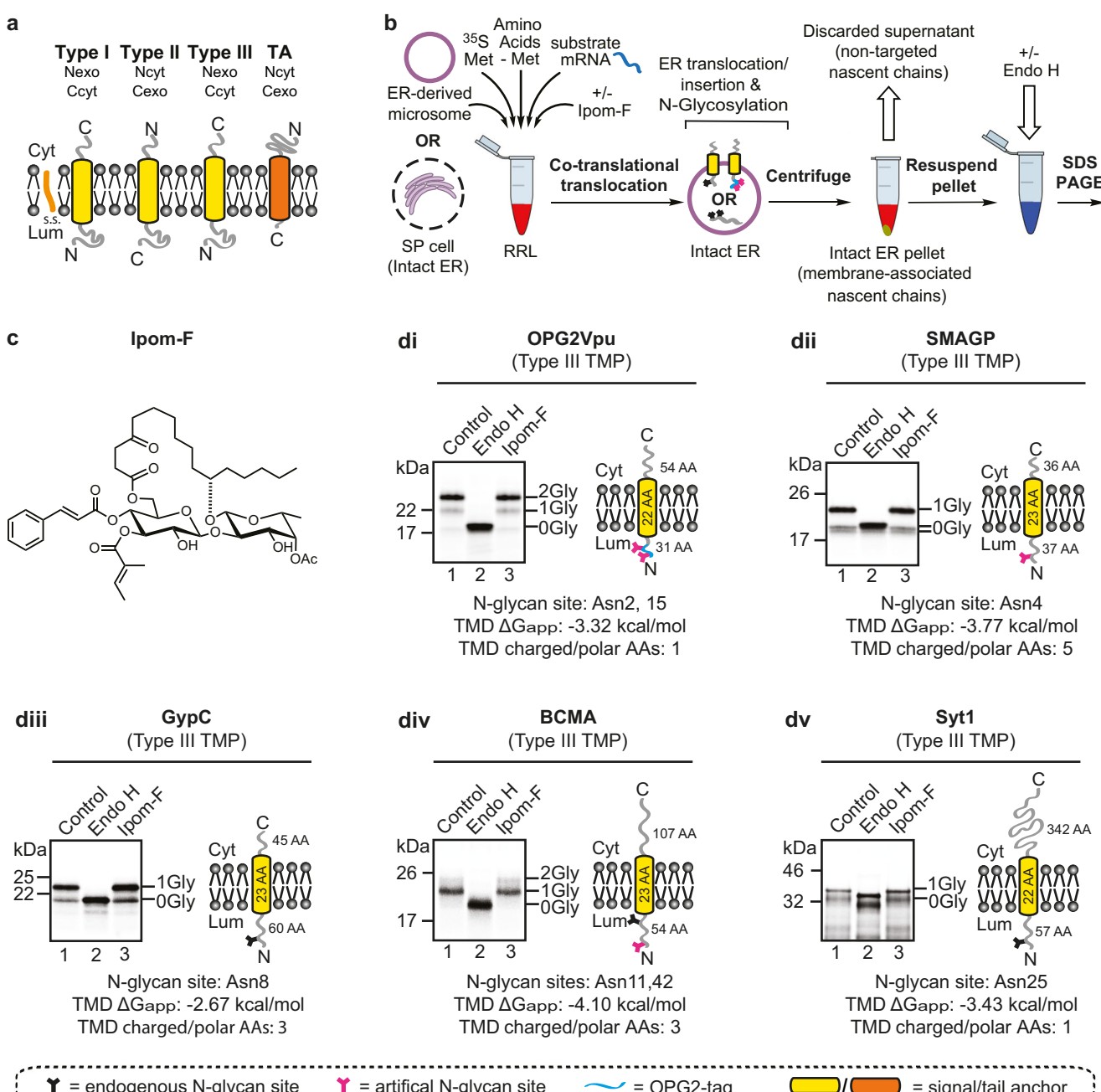

**Fig. 1 Type III TMPs are resistant to Ipom-F. a** Schematics of type I, type II, type III and tail-anchored (TA) proteins showing their topologies in the ER membrane. **b** Outline of the in vitro assay where either canine pancreatic microsomes or semi-permeabilised HeLa cells were used as sources of ER membrane; following translation, membrane inserted radiolabelled precursor proteins are recovered by centrifugation and analysed by SDS-PAGE and phosphorimaging. The N-glycosylation of lumenal domains, confirmed by treatment with endoglycosidase H (Endo H), indicates successful membrane translocation/insertion. **c** The chemical structure of ipomoeassin F (Ipom-F), a potent and selective inhibitor of Sec61-mediated protein translocation[1]. **d** Model type III TMPs used in this study were synthesised as outlined in (**b**) using canine pancreatic microsomes: (**di**) an N-terminally OPG2-tagged form of the HIV protein Vpu (OPG2Vpu), (**dii**) small cell adhesion glycoprotein containing an artificial N-glycosylation site (SMAGP), (**diii**) glycophorin C (GypC), (**div**) tumour necrosis factor receptor superfamily member 17 containing an artificial N-glycosylation site (BCMA) and (**dv**) synaptotagmin 1 (Syt1). Variably N-glycosylated (XGly) and non-glycosylated (0Gly) species are indicated, having been confirmed using Endo H (lane 2). Other symbols are: AAs, amino acid residues; Cyt, cytosol; Lum, ER lumen; RRL, rabbit reticulocyte lysate; s.s., N-terminal signal sequence. Estimated hydrophobicity values ($\Delta$Gapp)[42] of predicted transmembrane domains (TMDs) in kcal/mol were calculated using: http://dgpred.cbr.su.se/ (full protein scan option). Charged amino acid residues, D, E, H, K, R; polar amino acid residues, N, Q, S, T, Y.

the wider EMC (Fig. 2c)[10,14,16]. In contrast, a knock-down of Sec61α has no detectable effect on the three EMC subunits studied (Fig. 2c). We noted a lack of any obvious canine EMC5 homologue, and speculate that an apparent increase in EMC6 detected in dog pancreatic microsomes (versus sheep) may be

compensatory (Fig. 2c, lane 5 versus lane 6, Supplementary Fig. 2b and c).

We then investigated the effects of individual Sec61α, EMC2 or EMC5 knock-downs (71, 44 and 88% depletion respectively; Fig. 2d) on the in vitro ER insertion of different types of

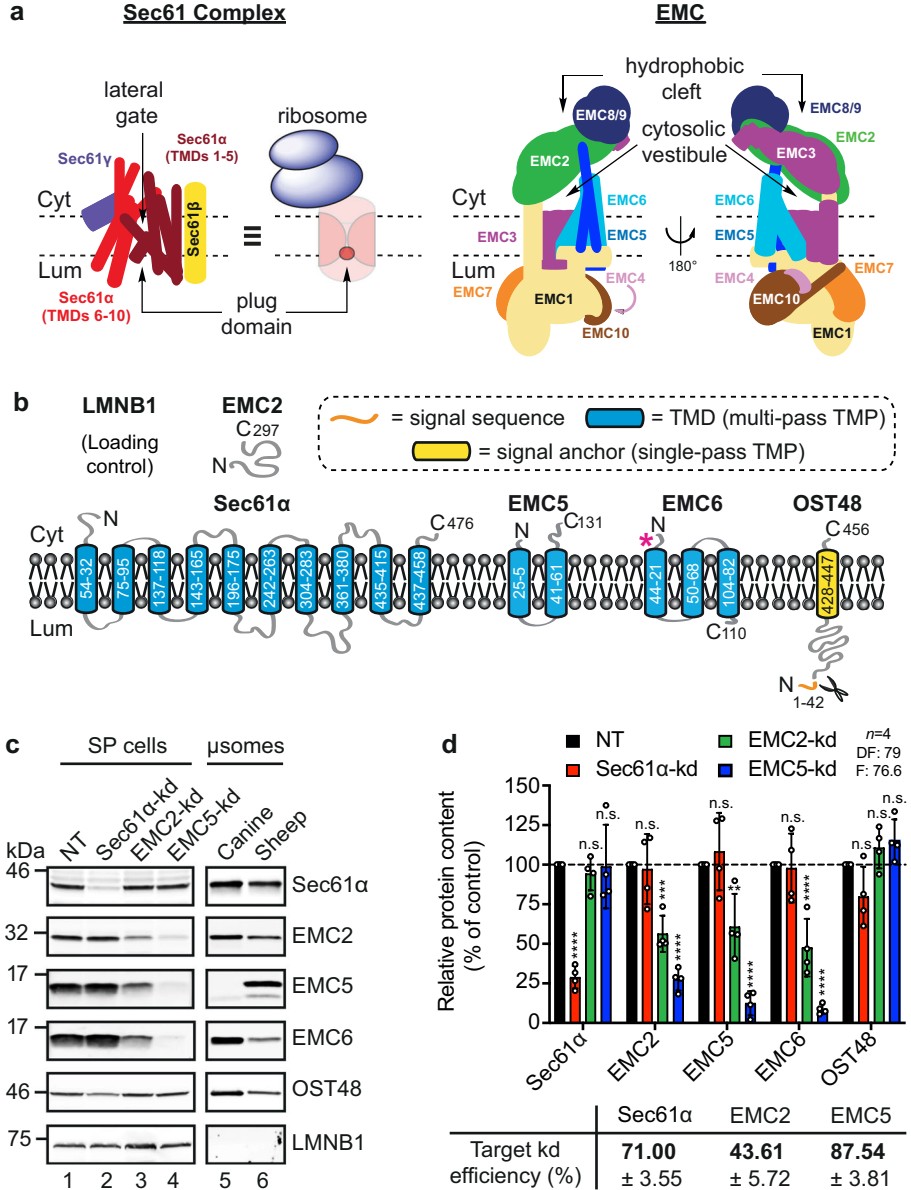

**Fig. 2 siRNA-mediated knock-down of Sec61α, EMC2 and EMC5. a** Schematics of the Sec61 complex and the EMC. The heterotrimeric Sec61 complex (α, β, γ subunits) forms a protein-conducting aqueous channel through which polypeptides are translocated into and across the ER membrane. Its actions are regulated via a plug domain and lateral gate[31,32]. The EMC is organised into a basket-shaped cytosolic domain (EMC2, EMC8/9), a transmembrane-spanning region (EMC3, EMC5, EMC6) and an L-shaped lumenal region (EMC1, EMC4, EMC7, EMC10). The insertase site is formed by EMC3/EMC6 near the cytosolic vestibule, whilst the hydrophobic cleft may have a role in client transmembrane domain capture[11–14]. **b** Outlines of targets for siRNA-mediated knock-down: Sec61α, EMC2, EMC5, together with additional proteins EMC6 and OST48 (48 kDa subunit of the oligosaccharyl-transferase complex) analysed by immunoblotting are shown. LMNB1, (Lamin-B1) was used as a loading control for the quantity of SP cells used in each experiment. TMD, transmembrane domain; TMP, transmembrane protein. **c** The effects of transfecting HeLa cells with non-targeting (NT; lane 1), Sec61α-targeting (lane 2), EMC2-targeting (lane 3) and EMC5-targeting (lane 4) siRNAs were determined after semi-permeabilisation by immunoblotting for Sec61α, EMC2, EMC5 and EMC6 subunits, OST48 and LMNB1. ER-derived microsomes (μsomes) of dog (lane 5) and sheep (lane 6) origin were immunoblotted for the same components. **d** The efficiencies of siRNA-mediated knock-down (bold) were calculated as a proportion of the signal intensity obtained with the NT control (set as 100%). Bar quantifications are given as means ± SEM for four separate siRNA treatments ($n = 4$, biologically independent experiments) with statistical significance of siRNA-mediated knock-downs (two-way ANOVA, DF and F values shown in the figure) determined using Tukey's multiple comparisons test. Statistical significance is given as n.s., non-significant $P > 0.1$; **$P < 0.01$; ***$P < 0.001$; ****$P < 0.0001$.

membrane proteins using SP cells depleted of each subunit (Fig. 3a)[28,34,35]. A knock-down of Sec61α resulted in a >60% reduction in the membrane integration of the type II TMP asialoglycoprotein receptor (ASGR1OPG2; Fig. 3b) and no further inhibition of ASGR1OPG2 insertion was achieved by adding Ipom-F (Fig. 3c, d), consistent with the previously reported Sec61-dependency of ASGR1[19,36,37] and Sec61α as the

primary target of Ipom-F[1]. As anticipated, knock-down of EMC5 had no effect on ASGR1OPG2 insertion, which remained sensitive to Ipom-F treatment (Fig. 3c, d). Unexpectedly, however, the modest knock-down of EMC2 (~44%) enhanced the membrane integration of ASGR1OPG2 by >90%, perhaps indicating an as yet undefined regulatory role for this subunit (Fig. 3c, d)[14].

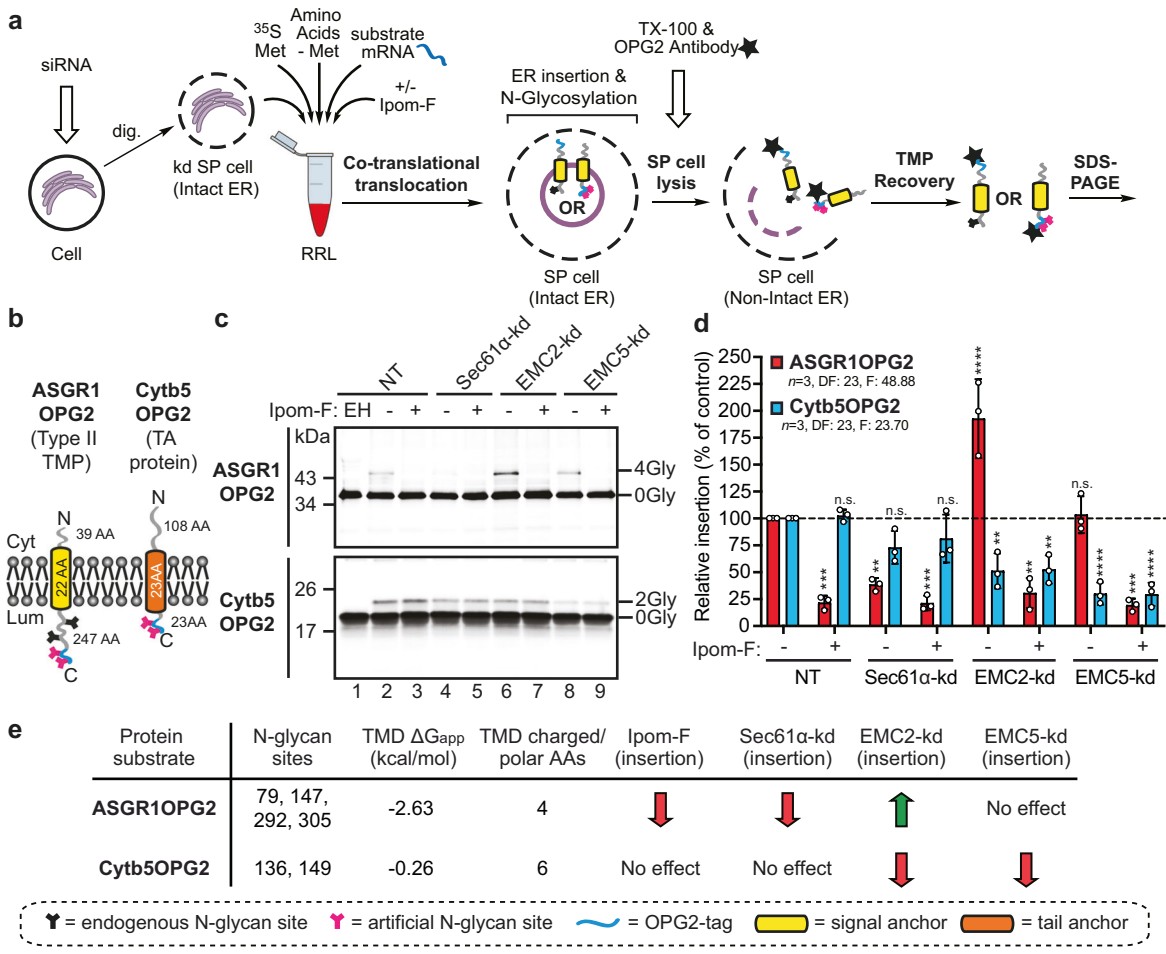

**Fig. 3 Sec61 and EMC knock-downs selectively inhibit TMP insertion. a** Schematic of the modified in vitro assay used for analysis of total translation products. Following translation, SP cells depleted of ER membrane components using siRNA were solubilised using Triton (TX-100). Total radiolabelled products (i.e. membrane-associated and non-targeted nascent chains) were recovered by immunoprecipitation via the OPG2-tag and analysed by SDS-PAGE and phosphorimaging. **b** Schematics of defined model substrates; the type II TMP asialoglycoprotein receptor 1 with an OPG2-tag (ASGR1OPG2) and the TA protein cytochrome b5 with an OPG2-tag (Cytb5OPG2). **c** ASGR1OPG2 (top panel) and Cytb5OPG2 (bottom panel) synthesised in the presence and absence of 1 μM Ipom-F using SP cells pre-treated with the indicated siRNA (NT, lanes 1–3; Sec61α-targeting, lanes 4–5; EMC2-targeting, lanes 6–7; EMC5-targeting, lanes 8–9) were recovered and analysed as described in (**a**). **d** Relative membrane insertion efficiencies were calculated using the ratio of N-glycosylated protein to non-glycosylated protein, relative to the NT control (set to 100% insertion efficiency). Quantifications are given as means ± SEM for independent insertion experiments from separate siRNA treatments performed in triplicate ($n = 3$, biologically independent experiments). The statistical significance of differences in the values for siRNA±Ipom-F translocation efficiency relative to the control (two-way ANOVA, DF and F values shown in the figure) were determined using Tukey's multiple comparisons test. **e** The effects of Ipom-F and Sec61α, EMC2 and EMC5 knock-downs on ASGR1OPG2 and Cytb5OPG2 insertion are summarised together with biochemical properties of their transmembrane domains (TMDs). Estimated hydrophobicity values (△Gapp)[42] and the number of charged/polar amino acid residues (AAs) in the transmembrane domain were calculated as described in the legend to Fig. 1. Statistical significance is given as n.s., non-significant $P > 0.1$; **$P < 0.01$; ***$P < 0.001$; ****$P < 0.0001$.

In contrast, the membrane insertion of the TA protein cytochrome b5 (Cytb5OPG2; Fig. 3b) was reduced by ~70% for the EMC5 knock-down and ~50% for the less-efficient EMC2 depletion, whilst its integration is not significantly impaired following knock-down of Sec61α or by the addition of Ipom-F (Fig. 3c, d). These data further support the view that Cytb5 can exploit multiple, often redundant, pathways of ER membrane insertion[5,18,38–41] and confirm the fidelity and efficiency of N-glycosylation for each knock-down condition even when key components of the N-glycosylation machinery, such as OST48[10,26], may be reduced following knock-down of Sec61α.

Taken together, these findings show that using a knock-down approach we can distinguish at least two different routes for the insertion of single-pass TMPs into the ER. The first, as exemplified by ASGR1, is sensitive to Ipom-F[1], utilises the Sec61 translocon[36] and is unaffected by depletion of EMC5[19] (Fig. 3e). The second, as illustrated by Cytb5, exhibits a strong

dependency on the EMC for membrane integration, but appears unaffected by loss of Sec61 function.

**Type III TMPs variably utilise the EMC as an ER membrane insertase.** Having incorporated an OPG2-tag into each of our model type III proteins and confirmed their continued resistance to Ipom-F (Supplementary Fig. 3, see also "Methods"), we proceeded to use SP cells to probe the potential role of the EMC during the insertion of type III TMPs by comparing the effects of EMC depletion with loss of Sec61α function through both siRNA-mediated knock-down and Ipom-F inhibition (Fig. 4). Consistent with their resistance to the inhibitory effects of Ipom-F, four of the five type III TMPs studied were unaffected by the depletion of Sec61α whether or not Ipom-F was also present (Fig. 4b, c, see: OPG2Vpu, SMAGPOPG2, BCMAOPG2 and Syt1OPG2). However, Sec61α depletion did result in a statistically significant

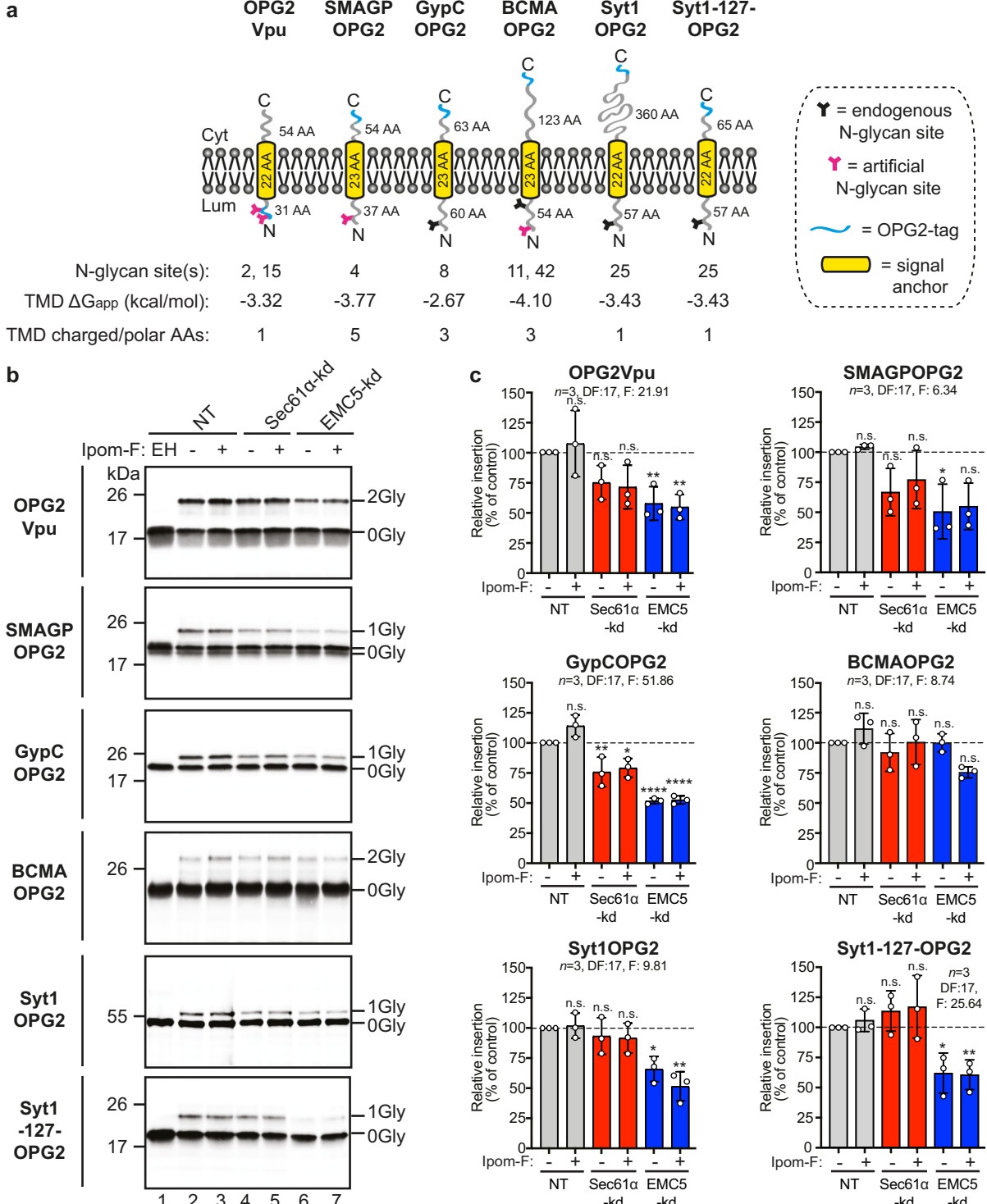

**Fig. 4 Type III TMPs utilise the EMC for membrane integration. a** Representative structures of type III TMPs (all OPG2-tagged as indicated) together with the biochemical properties of their transmembrane domains (TMDs). **b** Type III TMPs were synthesised using SP cells pre-treated with the indicated siRNA (NT, lanes 1–3; Sec61α-targeting, lanes 4–5; EMC5-targeting, lanes 6–7), recovered and analysed as described in the legend to Fig. 4. **c** Quantification of type III TMP insertion for HIV protein Vpu (OPG2Vpu), small cell adhesion glycoprotein containing an artificial N-glycan site (SMAGPOPG2), glycophorin C (GypCOPG2), tumour necrosis factor receptor superfamily member 17 containing an artificial N-glycan site (BCMAOPG2), synaptotagmin 1 (Syt1OPG2) and a cytosolically truncated version of Syt1OPG2 (Syt1-127-OPG2). Estimated hydrophobicity values ($\triangle$Gapp)[42] and the number of charged/polar amino acid residues (AAs) in the transmembrane domain are as shown in Fig. 1. Statistical significance (two-way ANOVA, DF and F values shown in the figure) was determined as indicated in the legend to Fig. 3. Statistical significance is given as n.s., non-significant $P > 0.1$; *$P < 0.05$; **$P < 0.01$; ****$P < 0.0001$.

(~24%) reduction in the membrane insertion of GypCOPG2, which we noted has the least hydrophobic transmembrane domain of the type III TMPs analysed (Fig. 4; $\Delta G_{app}$ of −2.67 kcal/mol versus −3.32 to −4.10 kcal /mol)[42].

The knock-down of EMC5 had a striking effect on the membrane integration of four OPG2-tagged model type III TMPs, with HIV-Vpu, GypC, SMAGP and Syt1 all showing significantly lower levels of membrane integration (34–50% reduced; Fig. 4b, c). Only the insertion of BCMA (Fig. 4b, c), a type III TMP bearing an extremely hydrophobic transmembrane domain (Fig. 4a; Supplementary Fig. 4), was unaffected. Thus, we conclude that the majority of type III proteins require a functional EMC for their efficient membrane insertion. Furthermore, and in contrast to most known EMC clients[11,19–21], this EMC-dependence does not appear to be strongly influenced by the relative polarity of the transmembrane domains present in our model type III TMPs (Fig. 4a, HIV-Vpu, GypC, SMAGP have 1, 3 and 5 hydrophilic residues respectively). However, by comparing full-length Syt1, which has a longer than the average cytosolic domain (Supplementary Fig. 4), with a C-terminally truncated version more typical of our other model proteins (Syt1-127-OPG2), we find evidence that the length of the cytosolic domain may influence the EMC dependency of some type III TMPs (Fig. 4, see "Discussion").

**Co-operation between Sec61 and the EMC for type III TMP insertion.** Having observed individual roles for both Sec61 and the EMC during GypC integration (Fig. 4), we investigated the potential interplay between these two complexes using co-depletion. Having established that double siRNA-mediated knock-downs of Sec61α + EMC2 and Sec61α + EMC5 subunits were of comparable efficiency to individual knock-downs (Fig. 5a, b versus Fig. 2c, d), we again utilised OPG2-tagged ASGR1 and Cytb5 (Fig. 5c) as benchmark TMPs. In accordance with its previously observed strong Sec61-dependence (Fig. 3), the reduction in ASGR1 insertion following double knock-down of both Sec61α + EMC2 and Sec61α + EMC5 was comparable to that achieved by Sec61α depletion alone. Interestingly, the increase in ASGR1 insertion observed after EMC2 depletion was lost when Sec61α was co-depleted (Fig. 5c, d versus Fig. 3c, d, see "Discussion"). Conversely, both double knock-downs resulted in a reduction of Cytb5 insertion that was less pronounced than that achieved by knocking down EMC5 alone and only marginally more effective than depleting Sec61α (Fig. 5c, d). We attribute this behaviour to the ability of Cytb5 to utilise several alternate, and apparently redundant, pathways for ER membrane insertion[5,41]. The Ipom-F sensitivity of ASGR1 or Cytb5 was unaltered by the double depletion of Sec61 and EMC subunits when compared to single subunit depletions (Fig. 5c, d versus Fig. 3c, d).

In the case of type III TMP integration, we saw a strong enhancement of the membrane insertion defect following double depletion of Sec61α + EMC5 for three model proteins, HIV-Vpu, SMAGP and GypC (65 to 75% reduction; Fig. 5). When compared to the effect of the individual knock-downs (Fig. 4 versus Fig. 5), this provides clear evidence for some element of mechanistically important co-ordination or synergy between the Sec61 complex and EMC during their membrane insertion. Even for BCMA, which was unaffected by the individual depletion of subunits from either complex, double subunit depletions now resulted in a reduction of membrane insertion by 36% (Fig. 5c, d).

In short, for the majority of type III TMPs tested, efficient membrane integration requires both the Sec61 complex and the EMC, in line with previous reconstitution studies that have implicated each complex in their membrane insertion[4,19]. Paradoxically, a double depletion of Sec61α + EMC5 reversed the ~34% reduction in the membrane insertion of Syt1 seen

following EMC5 depletion alone (Figs. 4 and 5). However, this effect was not seen with the truncated Syt1-127 variant (Figs. 4 and 5), further indicating that the comparatively long C-terminal region of Syt1 can influence its membrane insertion (see "Discussion").

**SRP receptor promotes insertion of type III TMPs.** To probe how the Sec61 complex and the EMC may co-operate during the integration of bona fide type III TMPs, we next investigated the role of the SRP receptor complex that is predicted to act upstream of these two membrane insertase complexes (Fig. 6a). To this end, we first analysed the levels of the SRP receptor α-subunit (SRα) in SP cells depleted of Sec61 and/or EMC sub-units. We found SRα expression was upregulated following Sec61α depletion (Sec61α-kd, Sec61α + EMC2-kd, Sec61α + EMC5-kd), as previously reported[26], but was either unaffected (EMC2-kd) or moderately reduced (EMC5-kd) by EMC subunit depletion alone (Supplementary Fig. 5). Furthermore, SP cell membrane-associated levels of SRP54, the SRP subunit that engages SRα[43], were not significantly altered by any knock-down condition used in this study (Supplementary Fig. 5). Hence, initial access to the Sec61 complex via the SRP-dependent co-translational pathway is most likely either unperturbed or enhanced at the level of SRP-mediated membrane targeting and SRP receptor binding following such knock-downs.

Having established that we could efficiently deplete SRα by siRNA-mediated knock-down, both alone and in combination with Sec61α or EMC5 subunits (~70– 90%; Fig. 6b, c), we investigated the effects of such depletions on membrane insertion. For the type II TMP, ASGR1, knock-down of SRα alone had a minor though non-significant effect on membrane integration (SRα-kd = 34% reduction versus Sec61α-kd = 62% reduction; see "Discussion"), whilst co-depletions of SRα did not enhance the effect of knocking down Sec61α or EMC5 alone (Fig. 6d, e). Likewise, for the TA protein, Cytb5, co-depletion of SRα did not enhance the already strong membrane insertion defect observed upon EMC5 knock-down, although we did observe an enhanced defect in Cytb5 insertion when SRα and Sec61α knock-downs were combined (Fig. 6d, e).

Type III TMPs that showed the strongest defects in membrane insertion following combined knock-down of EMC5 + Sec61α (Vpu and GypC; see Figs. 5f and 6) also showed significantly reduced levels of membrane insertion following SRα depletion alone (Fig. 6e). When SRα depletion was combined with that of either EMC5 or Sec61α, an even stronger defect in the membrane insertion of Vpu and GypC was observed, comparable to the reduction achieved following co-depletion of EMC5 and Sec61α (Fig. 6e). This behaviour is consistent with a model for type III TMP insertion where the SRP receptor, EMC and Sec61 complex all participate in a single co-translational pathway (Fig. 7). Comparable, though less pronounced, defects in the membrane insertion of BCMA and Syt1-127 were also seen after combined depletions (EMC5 + Sec61α-kd, SRα + EMC5-kd and SRα + Sec61α-kd; see Fig. 6e), further supporting the proposal that these three components act on the same pathway. Likewise, for Syt1, where co-depletion of EMC5 and Sec61α reversed the insertion defect seen upon EMC5 knock-down alone (cf. Figs. 4 and 5), co-depletion of SRα with either EMC5 or Sec61α resulted in an enhanced insertion defect. Only in the case of SMAGP and Syt1-127 did a knock-down of EMC5 and Sec61α appear more effective than either of the SRα combinations tested (Fig. 6e). In summary, these data strongly suggest that the model type III TMPs investigated in this study are targeted to their site of ER membrane integration via the SRP-dependent pathway, although this ER delivery route may be non-exclusive (see "Discussion").

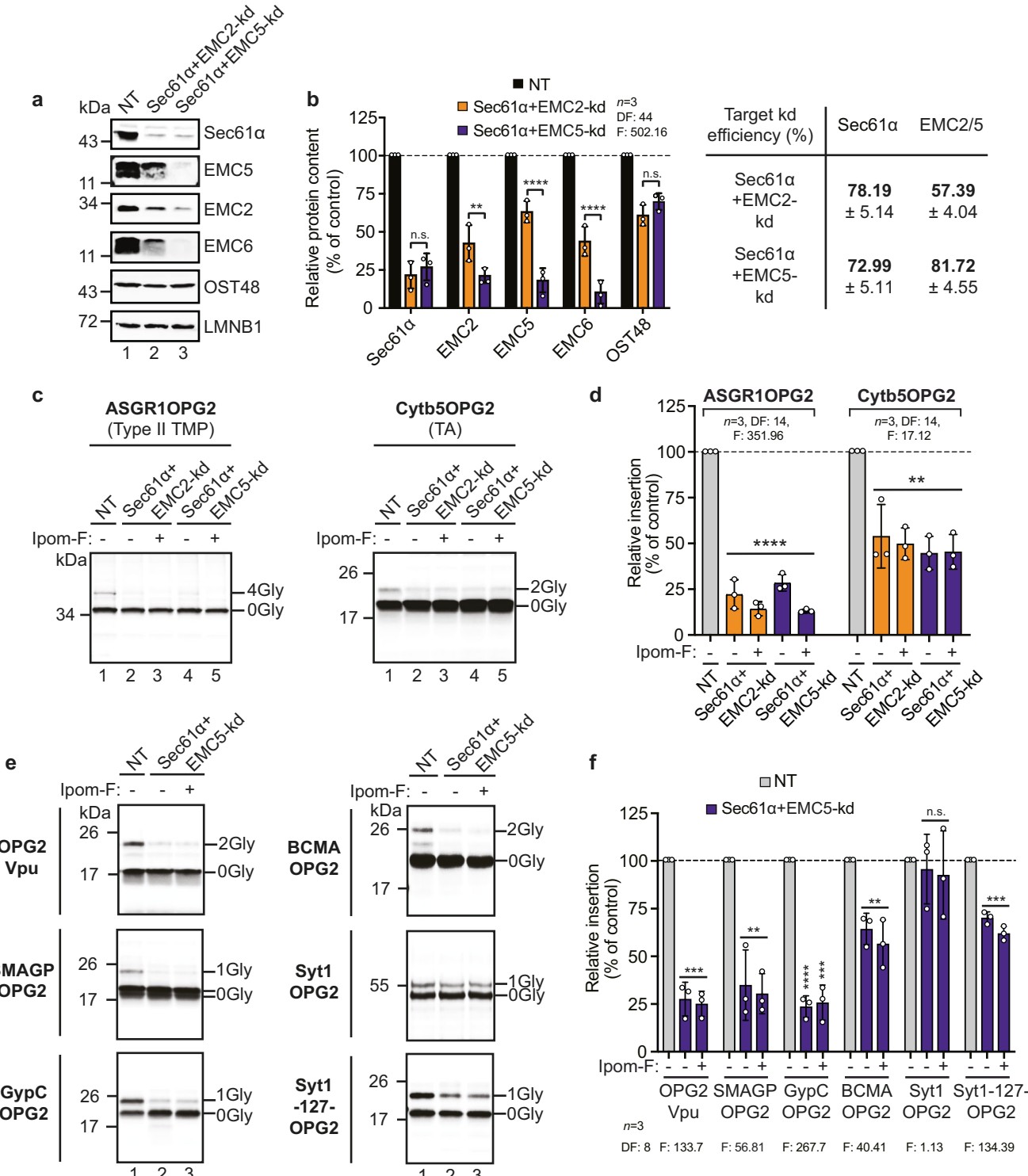

## Discussion

Although previously implicated in the biogenesis of specific TA proteins and subdomains of multi-pass TMPs[14,16–19], the importance of EMC-mediated membrane insertion for the biogenesis of most single-pass TMPs remains under-explored. Prompted by the ability of several naturally occurring type III TMPs (four mammalian and one viral) to completely bypass the effects of Sec61 inhibitors (Fig. 1 and Supplementary Fig. 1)[1], we have used Ipom-F (Figs. 1, 3–5) and selective siRNA-mediated knock-downs of Sec61, EMC and SRP receptor complex subunits,

both alone and in combination (Figs. 2–6), to explore the biogenesis of type III TMPs and provide a unifying model for their ER insertion (Fig. 7).

Amongst our panel of type III TMPs, only GypC showed a significant reduction in membrane integration following Sec61α depletion (~25%, Fig. 4). Thus, loss of Sec61 function alone, either through knock-down or Ipom-F inhibition, has little or no effect on the type III TMPs tested. A much clearer disruption is seen following EMC5 knock-down, with four type III TMPs showing substantially reduced membrane insertion (~34–50% lower; Fig. 4);

**Fig. 5 Depletion of Sec61 and EMC further inhibits type III TMP insertion. a** The effects of transfecting HeLa cells with non-targeting (NT; lane 1), Sec61α +EMC2-targeting (lane 2) and Sec61α+EMC5-targeting (lane 3) siRNAs were determined after semi-permeabilisation by immunoblotting for Sec61α, EMC2, EMC5 and EMC6 subunits, OST48 and LMNB1. **b** The efficiencies of siRNA-mediated knock-down (bold) were calculated as a proportion of the signal intensity obtained with the NT control (set as 100%). **c** ASGR1OPG2 (left panel) and Cytb5OPG2 (right panel) synthesised in the presence and absence of 1 μM Ipom-F using SP cells pre-treated with the indicated siRNA (NT, lanes 1–3; Sec61α + EMC2-targeting, lanes 4–5; Sec61α + EMC5-targeting, lanes 6–7) were recovered and analysed as described in the legend to Fig. 3. **d** The relative membrane insertion efficiencies for ASGR1OPG2 and Cytb5OPG2 in SP cells depleted of Sec61α + EMC2 or Sec61α + EMC5 subunits (double knock-downs) were calculated using the ratio of N-glycosylated protein to non-glycosylated protein, relative to the NT control (set to 100% insertion efficiency). **e** Type III proteins (OPG2-tagged) as detailed in the legend to Fig. 4 were synthesised in the presence and absence of 1 μM Ipom-F using SP cells pre-treated with the indicated siRNA (NT, lane 1; Sec61α + EMC5-targeting, lanes 2–3), recovered, analysed and (**f**) their relative membrane-insertion efficiencies quantified as described in parts (**c**) and (**d**). All quantifications are given as means ± SEM for three separate siRNA treatments (*n* = 3 biologically independent experiments). Statistical significance (two-way ANOVA, DF and F values shown in the figure) was determined using Tukey's multiple comparisons test and is given as n.s., non-significant >0.1; **$P$ < 0.01; ***$P$ < 0.001; ****$P$ < 0.0001.

thereby confirming that the EMC plays an important role in the biogenesis of bona fide type III TMPs[19]. However, when knock-downs of Sec61α and EMC5 are combined, we see clear evidence that both complexes are necessary for efficient type III TMP insertion (Fig. 5). Furthermore, type III TMPs which show the biggest reductions in membrane insertion following EMC5 depletion alone (HIV-Vpu, SMAGP and GypC; Fig. 4) all showed clearly enhanced defects following a double knock-down with Sec61α (~65 to ~75% reduction; Fig. 5). We therefore conclude that the biogenesis of type III TMPs normally involves both the EMC and the Sec61 complex (cf. Fig. 7).

The co-translational insertion of nascent membrane proteins via the Sec61 complex is typically dependent upon their SRP-mediated delivery (cf. Fig. 6a), and we therefore tested the hypothesis that the SRP receptor acts prior to the EMC/Sec61-dependent membrane insertion of type III TMPs. With the exception of the viral Vpu protein, the depletion of SRα alone had a comparatively modest effect on all three classes of TMP analysed (Fig. 6e). Although we cannot exclude that the residual SRα remaining after depletion (~30%) can service multiple ER membrane insertion sites, this seems unlikely given SRP receptor levels are rate limiting for SRP-dependent targeting[44]. Hence, we speculate that the comparatively modest defects seen following SRα depletion alone reflect the ability of nascent TMPs to exploit alternative mechanisms for ER delivery, such as the putative mammalian SND pathway[5,41,45,46]. For type III TMPs, the impact of SRα depletion is clearer when combined with the knockdown of EMC5 or Sec61α. In each case, the co-depletion of SRα + EMC5, SRα + Sec61α or EMC5 + Sec61α all result in strong and comparable defects in membrane insertion (cf. Fig. 6e), leading us to conclude that the SRP receptor, EMC and Sec61 can all act in concert during type III TMP insertion (Fig. 7).

On the basis of these findings, we propose that, following SRP-mediated delivery to the ER localised SRP receptor[43] (Fig. 7a), the EMC and Sec61 complex are co-ordinated to provide an alternative pathway for the N-terminal translocation and membrane insertion of type III TMPs. Given the consistent insensitivity of type III TMPs to inhibition by Ipom-F (Figs. 1 and 3–5), it is extremely unlikely that this route involves membrane access via the Sec61 lateral gate, which would probably be occluded by bound Ipom-F[1] (Fig. 7b, route 1), as recently established for mycolactone[47]. Thus, rather than directly facilitating the integration of type III TMPs, we propose that the Sec61 complex acts to complement the membrane insertase activity of the EMC (Fig. 7b, route 2). In practice, such a translocation independent role for the Sec61 complex may involve its co-ordination of SRP-dependent delivery[48] and/or binding to the ribosome-nascent chain complex[31]. Intriguingly, we find preliminary evidence that the EMC may also influence co-translational insertion via the Sec61 complex. Hence, depletion of EMC2, but not EMC5,

selectively enhanced ASGR1 insertion by ~90% via a pathway that remained Ipom-F sensitive (Fig. 3). This increase was not sustained following co-depletion of Sec61α and EMC2 (Fig. 5), further suggesting some interplay between Sec61 and the EMC (Fig. 7b, route 2) and supporting the proposal that the functions of the EMC extend beyond those of a simple membrane insertase[13].

Recent structural studies identified a conserved vestibule formed by EMC3/EMC6 as the site of EMC-mediated transmembrane domain insertion into the ER[11–14]. EMC3 is structurally homologous with YidC[13], an insertase that acts downstream of SRP in bacteria and facilitates membrane insertion, either alone or together with SecY (a Sec61α orthologue)[49–52]. Transient contact is observed between SecY and flexible cytosolic domains of YidC during the membrane insertion of substrates[50] and it was recently suggested that a YidC-SecYEG complex enables the insertion of certain bacterial membrane proteins[52]. By analogy, and whether transiently or via the formation of an as yet unidentified EMC-Sec61 complex, it seems feasible that the equivalent methionine-rich C1 loop and/or C-terminus of EMC3 may encounter Sec61α during the selective capture of type III TMP clients and direct their transmembrane domains to the EMC insertase site[13] (cf. Fig. 7b, route 2). In addition, as for YidC[51,53], positively charged regions within the cytosolic domains of one or more EMC subunits may promote binding to the ribosome-nascent chain complex (Fig. 7b, routes 2 and 3). Given that a stable or prolonged interaction between the EMC and a ribosome-engaged Sec61 complex operating on the normal co-translational pathway appears unlikely[12], the capacity of the EMC to support type III TMP insertion in isolation (Fig. 7b, route 3) is supported by the minimal defect observed following knock-down of Sec61α alone (Fig. 4).

The type III TMPs studied here are fairly typical of the wider group (Supplementary Fig. 4) and, on that basis, we propose that most type III TMPs (38 in humans, see Supplementary Data 1) follow our proposed biosynthetic pathway (Fig. 7). In contrast to previous studies of other clients, we find no evidence that transmembrane domains of enriched polarity[20,21] confer EMC-dependence to single-pass type III TMPs (Fig. 4, Fig. 6, Supplementary Fig. 4). Likewise, the transmembrane domains of the type III TMPs studied here (ΔG$_{app}$ of −2.67 to −4.10 kcal/mol; see Fig. 4), and the properties of type III SARS-CoV-2 envelope protein[54], indicate that low to moderate transmembrane domain hydrophobicity is also not a pre-requisite for EMC-dependent membrane insertion[15]. In the case of bona fide type III TMPs, we propose that it is the necessity to translocate their pre-formed hydrophilic N-terminus in concert with the integration of their single transmembrane domain[12] that dictates their entry into the EMC/Sec61-mediated membrane insertion pathway (Fig. 7).

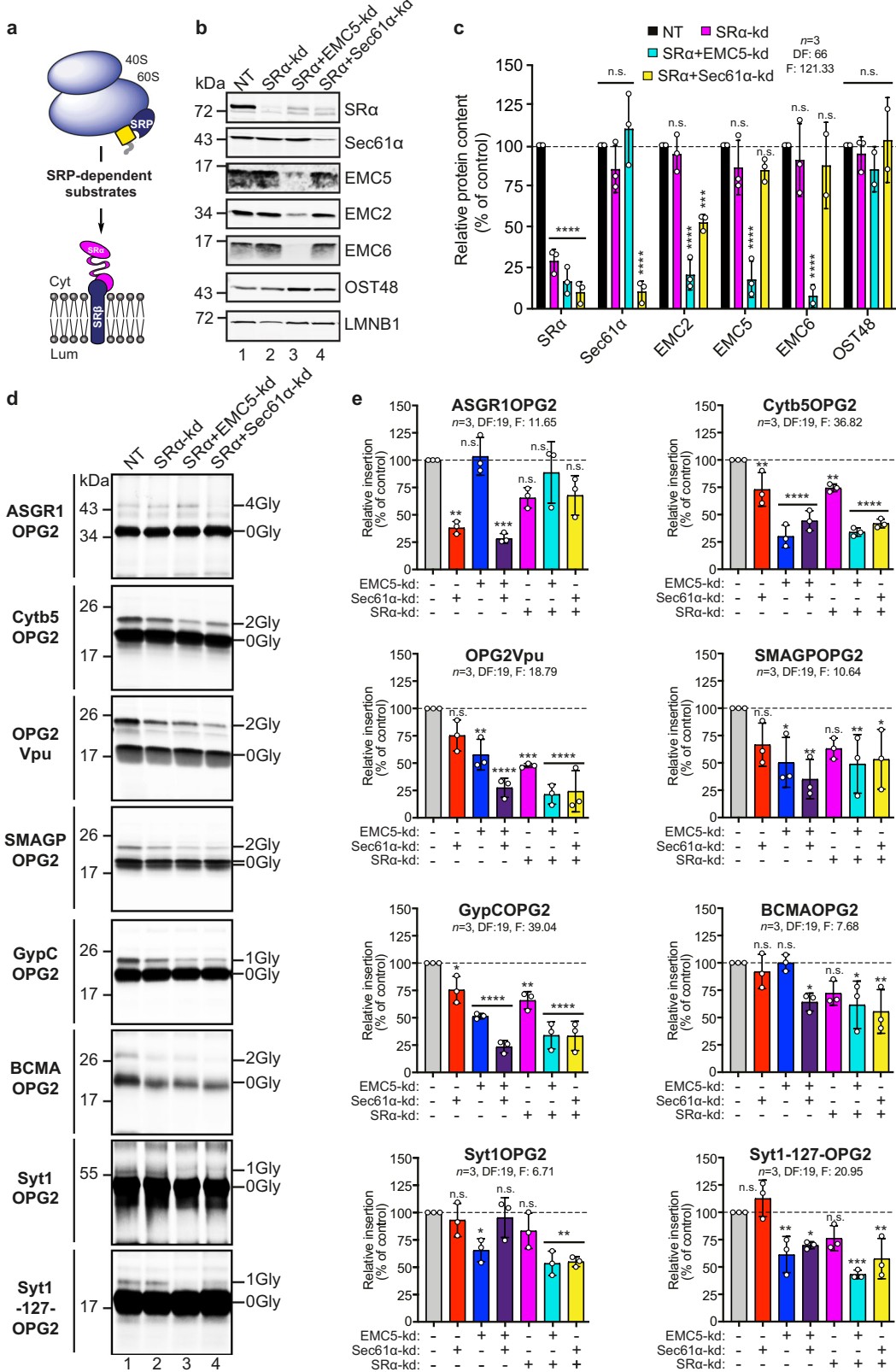

Our findings with BCMA and Syt1 indicate that individual variations in transmembrane domain hydrophobicity and cytosolic domain length may affect the efficiency of type III TMP insertion by this alternate EMC/Sec61-mediated route, perhaps by influencing transmembrane domain capture by and/or membrane insertion at the EMC[16,19]. In the case of BCMA ($\Delta G_{app}$ of

−4.10 kcal/mol), co-depletion experiments suggest at least some nascent chains follow the pathway depicted in Fig. 7. However, it also appears that BCMA can either utilise the residual SRP receptor/EMC/Sec61 machinery particularly effectively and/or exploit alternative, partially redundant, routes for its ER membrane insertion[5,18,41]. Type III TMPs with longer than average

**Fig. 6 Type III TMP insertion is dependent on the SRP receptor. a** A model of SRP-dependent targeting to the ER membrane: the SRP 'scans' the emerging nascent chain of a translating ribosome for hydrophobic ER signal-sequences or transmembrane domains, binds the ribosome-nascent chain complex and delivers it to the ER membrane via its interaction with the SRP receptor, prior to membrane insertion[43]. The SRP receptor is a heterodimer comprised of the ER membrane-integrated beta subunit (SRβ) anchoring the peripherally associated cytosolic alpha subunit (SRα). **b** The effect of transfecting HeLa cells with non-targeting (NT; lane 1), SRα-targeting (lane 2), SRα + EMC5-targeting (lane 3) and SRα + Sec61α-targeting (lane 4) siRNAs was determined after semi-permeabilisation by quantitatively immunoblotting for: SRα, Sec61α, EMC2, EMC5, EMC6, OST48 and LMNB1. **c** The efficiency and statistical significance of siRNA-mediated single and double SRα knock-downs were calculated as a proportion of the signal intensity obtained with the NT control (set as 100%), and using LMNB1 as a loading control, as before (see Fig. 2). Other components tested were unaffected (see Supplementary Fig. 5). **d** OPG2-tagged control (first and second panels) and type III TMPs synthesised using SP cells pre-treated with the indicated siRNAs (NT, lane 1; SRα lane 2, SRα + EMC5, lane 3; SRα + Sec61α-targeting, lane 4) were recovered and analysed as described in the legend to Fig. 3. **e** Relative membrane insertion efficiencies were calculated using the ratio of N-glycosylated protein to non-glycosylated protein, relative to the NT control (set to 100% insertion efficiency). To permit visual comparison across knock-down conditions, data are shown side by side with the membrane insertion efficiency achieved in SP cells depleted of either Sec61α and EMC5 subunits alone and in combination (these data are respectively from Figs. 3–5). Quantifications are given as means ± SEM for three separate siRNA treatments ($n = 3$, biologically independent experiments) and statistical significance (RM one-way ANOVA, DF and F values shown in the figure) was determined using Dunnett's multiple comparisons test (versus two-way ANOVA in Figs. 3–5). Statistical significance is given as n.s., non-significant $P > 0.1$; *$P < 0.05$; **$P < 0.01$; ***$P < 0.001$; ****$P < 0.0001$. N.B. When the data are re-analysed in this way, the effect of a single knock-down of Sec61α on the membrane insertion of Cytb5 is now deemed statistically significant (cf. Fig. 3).

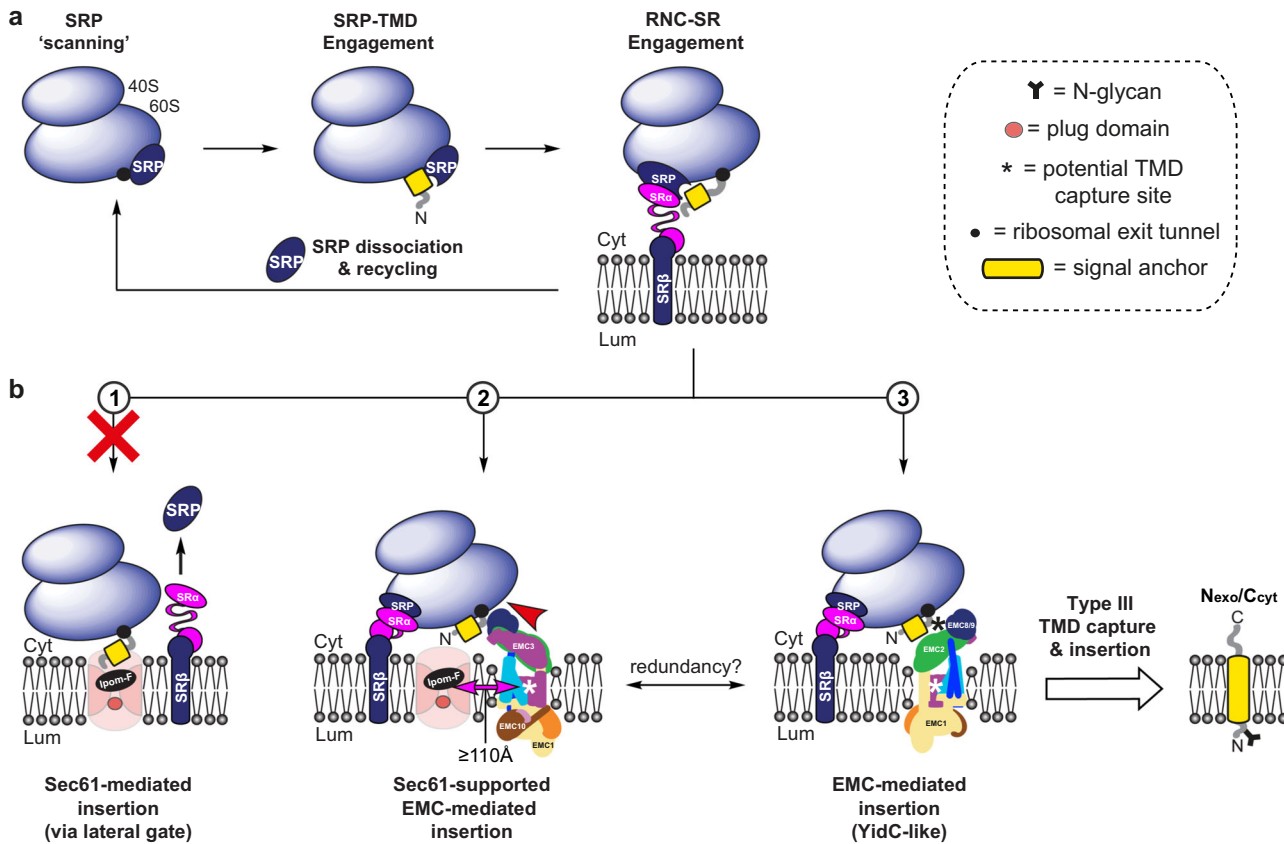

**Fig. 7 A model for type III TMP biogenesis. a** Type III TMPs are first delivered to the ER membrane via the SRP-dependent co-translational pathway (as in Fig. 6). **b** Following engagement of SRP-SRα, we propose that the Sec61 complex may act as a ribosome docking site but, as opposed to the insertion of type I/II TMPs which proceeds via the lateral gate (route 1), Sec61 does not fully engage the transmembrane domain. Instead, the type III transmembrane domain is captured by the EMC and inserted into the membrane, most likely via the EMC3/EMC6 insertase site (route 2; pink and cyan coloured membrane-spanning subunits respectively); an action assisted by local thinning of the membrane[13]. This mechanism is not impacted by the binding of Ipom-F which, by analogy to mycoclactone[47], impedes access to the Sec61 lateral gate. Alternatively, the Sec61 complex may support EMC-dependent translocation by facilitating the release of the SRP from the RNC-SRP-SR complex[36], thereby permitting type III transmembrane domain capture by the EMC without ribosomal engagement of the Sec61 complex. Whilst the relative orientations of Sec61, the EMC and the ribosome-nascent chain complex are as yet unknown, such a mechanism would be consistent with the recent suggestion that steric limitations (see red arrow head) between a Sec61-engaged ribosome and the cytosolic domain of the EMC restrict the proximity of the Sec61 lateral gate and the EMC insertase site to ~110Å[12] (cf. double headed pink arrow). Alternatively, and/or additionally, the EMC may bind directly to ribosomes and insert type III transmembrane domains independently of the Sec61 complex, in a manner analogous to that of YidC (route 3). EMC schematics for route 2 and 3 are shown rotated 180° around the vertical axis in order to show potentially relevant EMC subunit locations. Potential sites of client transmembrane domain capture by the EMC are indicated, these may include: (i) a flexible region of the EMC3/EMC6 insertase site (routes 2–3, white asterisk) and/or (ii) a hydrophobic cleft at the interface between EMC2-EMC8/9 (route 3, black asterisk).

**Table 1 Primer list 1.**

| New cDNA | Cloned cDNA | Vector | Forward primer (5'-3') | Reverse primer (5'-3') |
|---|---|---|---|---|
| ASGR1-OPG2 | ASGR1[59] | SP64 | AACGGAACAGAAGGACCAAACTTCTACGTACCA TTCAGCAACAAACAGGCTAATAATTTATTTCTT CAATGCCTCGACCTGCC | CTATTAGCCTGTGTTTTGTTGCTGAATGGTACGTAGAAGTTTGGTC CTTCTGTTCCGTTAAGGAGAGGTGGCCTCCTGGCCTTGTC |
| BCMA-Y13T | BCMA | pCMV3 | GTGCTCCCAAAATGAAACTTTTGACAGTTTGTTGC | GCAACAAACTGTCAAAAGTTTCATTTTGGAGCAC |
| BCMA-Y13T-OPG2 | BCMA-Y13T | pCMV3 | AACGGAACAGAAGGACCAAACTTCTACGTACCAT TCAGCAACAAACAGGCTAATAAATCTAGAGCGG CCGCCGAATTCGGGCC | CTATTAGCCTGTGTTTTGTTGCTGAATGGTACGTAGAAGTTTGGTC CTTCTGTTCCGTTCCTAGCACAGAAATTGATTTCTCTATC |
| GypC-OPG2 | GypC[4] | pGEM4 | AACGGAACAGAAGGACCAAACTTCTACGTACCAT TCAGCAACAAACAGGCTAATGAGGGACAACAGA CTTCACTTCCCTGAATG | CTATTAGCCTGTGTTTTGTTGCTGAATGGTACGTAGAAGTTTGGTC CTTCTGTTCCGTTAATAAAGTACTCCTTTCTGCTGCTATCACC |
| SMAGP-L4N | SMAGP | pCMV3 | GGTACCATGACCAGCAACCTGACTACTCCT | AGGAGTAGTCAGGTTGCTGGTCATGGTACC |
| SMAGP-L4N-OPG2 | SMAGP-P4N | pCMV3 | AACGGAACAGAAGGACCAAACTTCTACGTACCATT CAGCAACAAACAGGCTAATGAATCTAGAGGCGGC CGCCGAATTCGGGCC | CTATTAGCCTGTGTTTTGTTGCTGAATGGTACGTAGAAGTTTGGT CCTTCTGTTCCGTTGATGAAATATTCCTCTTTCTGCTG |
| Syt1-OPG2 | Syt1[1] | pcDNA3 | AACGGAACAGAAGGACCAAACTTCTACGTACCATT CAGCAACAAACAGGCTAATAAGGATCCACTAGTC CAGTGTGGTGGAATTC | CTATTAGCCTGTGTTTTGTTGCTGAATGGTACGTAGAAGTTTG GTCCTTCTGTTCCGTTCCTTCTTGACAGCCAGCATGGCATCAACCTCC |
| Syt1-127-OPG2-3M | Syt1[1] | pcDNA3 | AACGGAACAGAAGGACCAAACTTCTACGTACCATT CAGCAACAAACAGGCTAATGATGTAAACTGATG GAGAGAAAGGAAGAGC | CTATTACACATCATCATGCCTGTTTTGTTGCTGAATGGTACGTAGAA GTTTGGTCCTTCTGTTCCGTTCCGTTCCGGTTCAGCATCGTCATC |

A list of primer sequences that were used to generate new cDNAs used in this study by site-directed mutagenesis.

cytoplasmic domains (average and median length of 330 and 261 amino acids respectively; see Supplementary Fig. 4 and Supplementary Data 1), as exemplified by Syt1, appear resistant to co-depletion of EMC/Sec61 subunits, but do show a defect when either component is co-depleted with SRα (Figs. 5f, 6e). We speculate that the synthesis of the extra ~300 residues present in full-length Syt1 increases the residence time of its nascent chain on the ER membrane, thereby enhancing its ability to engage the residual components remaining after knockdown, and/or utilise alternative factors. In summary, we have identified key components that mediate type III TMP biogenesis and propose a unifying model whereby, following SRP-mediated delivery to the ER, these proteins are integrated into the membrane by the EMC acting in concert with the Sec61 complex (Fig. 7).

## Methods

**Ipom-F and antibodies**. Ipom-F was synthesised as previously described[55–57]. The goat polyclonal anti-LMNB1 antibody, which served as a loading control for the amount of SP cells present per translation reaction, was purchased from Santa Cruz (clone M-20, sc-6217). Rabbit polyclonal antibodies against EMC5 and EMC6 were purchased from Bethyl Laboratories (A305-832-A) and Abcam (ab84902) respectively. The mouse monoclonal antibody against EMC2 was purchased from Santa Cruz (sc-166011). The mouse monoclonal antibody recognising the OPG2-tag[22], rabbit polyclonal anti-OST48[28] and guinea pig anti-CAML[5] were as described previously. Rabbit antisera against Sec61α (C-terminus) and hSnd2 were gifts from Sven Lang (University of Saarland, Homburg, Germany) and rabbit antisera against SRα were provided by Martin Pool (University of Manchester).

**Constructs**. All cDNAs encode human genes unless specified otherwise. The cDNAs for BCMA (Uniprot: Q02223) and SMAGP (Uniprot: Q0VAQ4) were purchased from Sino Biological (HG10620-UT and HG25714-UT). The cDNA for the human immunodeficiency virus type I group M subtype B (isolate BRU/LAI) protein HIV-Vpu (Uniprot: P05923) was custom synthesised to order (GenScript)[58]. ASGR1[59], CD3δ[1], Cytb5OPG2[22], GypC[4], Ii[60], bovine PPL[23], yeast ppαF[22], Sec61βOPG2[22], rat Syt1[1], VCAM1[1] were as previously described. Artificial N-glycosylation sites (SMAGP-L4N, BCMA-Y13T) and OPG2-tagged substrates (ASGR1OPG2, SMAGP-L4N-OPG2, GypCOPG2, BCMA-Y13T-OPG2, Syt1OPG2, OPG2Vpu) were generated by site-directed mutagenesis (Stratagene QuikChange, Agilent Technologies) using the relevant forward and reverse primers (Integrated DNA Technologies). Linear DNA templates were generated by PCR and mRNA transcribed using T7 or SP6 polymerase. All primer combinations for mutagenesis and PCR are listed in Tables 1 and 2 respectively.

**Knock-down and preparation of SP cells**. HeLa cells (human epithelial cervix carcinoma cells, mycoplasma-free), as previously described[61], were provided by Martin Lowe (University of Manchester) and were cultured in DMEM supplemented with 10% (v/v) FBS and maintained in a 5% $CO_2$ humidified incubator at 37 °C. Knock-down of target genes were performed similarly to previously described methods[18,26,33]. Briefly, $1 \times 10^6$ cells were seeded per 10 $cm^2$ dish and, 24 h after plating, cells were transfected with either control siRNA (ON-TARGETplus Non-targeting control pool; Dharmacon), SEC61A1 siRNA (Sec61α-kd, GE Healthcare, sequence AACACUGAAAUGUCUACGUUUUU), TTC35 siRNA (EMC2-kd, Santa Cruz, sc-77588), MMGT1 siRNA (EMC5-kd, ThermoFisher Scientific, s41129) or SRPRA siRNA (SRα-kd, GE Healthcare, sequence GAG-CUUGAGUCGUGAAGACUU) at a final concentration of 20 nM using INTER-FERin (Polyplus, 409-10) as described by the manufacturer. For SEC61A1 silencing, either alone or in combination with MMGT1 silencing (Sec61α + EMC5-kd), the medium of the non-targeting control and targeted knock-downs was changed 24 h post-transfection and cells were re-transfected with siRNA a second time[26]. 96 h post-initial transfection, cells were SP using a modification of the previously described method[28]. Briefly, cells were detached by incubation with 3 mL of 0.25% trypsin-EDTA solution (Sigma-Aldrich) for 10 min at RT, mixed with 4 mL of KHM buffer (110 mM KOAc, 2 mM Mg(OAc)$_2$, 20 mM HEPES-KOH pH 7.2) supplemented with 100 μg/mL Soybean trypsin inhibitor (Sigma-Aldrich, T6522) and centrifuged at 250 **g** for 3 min at 4 °C. The pellet was resuspended in 4 mL KHM buffer supplemented with 80 μg/mL high purity digitonin (Calbiochem) and cells incubated on ice to selectively permeabilise the plasma membrane. After 10 min, cells were again centrifuged at $250 \times g$ for 3 min before resuspension in HEPES buffer (90 mM HEPES, 50 mM KOAc, pH 7.2) and incubation on ice for 10 min. Cells were pelleted by centrifugation once more, resuspended in 100 μL KHM buffer and endogenous mRNA removed by treatment with 0.2 U Nuclease S7 Micrococcal nuclease, from *Staphylococcus aureus* (Sigma-Aldrich, 10107921001) and 1 mM $CaCl_2$ at RT for 12 min before quenching by the addition of EGTA to 4 mM final concentration. Cells were centrifuged at $13,000 \times g$ for 1 min and resuspended in an appropriate volume of KHM buffer to give a suspension of $3 \times 10^6$ SP cells/mL as determined by trypan blue staining (Sigma-Aldrich, T8154). SP

**Table 2 Primer list 2.**

| Recombinant cDNA | Vector | Species | Forward primer (5′-3′) | Reverse primer (5′-3′) | RNA Polymerase |
|---|---|---|---|---|---|
| ASGR1[59] | SP64 | Human | CCAGAAACTCAGAAGGGTCG | CAGGAAACAGCTATGAC | SP6 |
| ASGR1-OPG2 | SP64 | | | | |
| BCMA-Y13T | pCMV3 | Human | CGCAAATGGGCGGTAGGCGTG | TAGAAGGCACAGTCGAGG | T7 |
| BCMA Y13T-OPG2 | pCMV3 | | | | |
| CD3δ[1] | pcDNA3 | Human | CGCAAATGGGCGGTAGGCGTG | TCACATCATCATCTTGTTCC GAGCCCAGTTTCC | T7 |
| Cytb5OPG2 | pcDNA3 | Human | CGCAAATGGGCGGTAGGCGTG | TAGAAGGCACAGTCGAGG | T7 |
| GypC[4] | pGEM4 | Human | GTGGATAACCGTATTACCGCC | CTCTGACGGCAGTTTACGAG | T7 |
| GypC-OPG2 | pGEM4 | | | | |
| Ii[60] | pGEM3 | Human | CTCTGACGGCAGTTTACGAG | GTGGATAACCGTATTACCGCC | SP6 |
| OPG2Vpu[58] | pTNT | HIV | GGGAAACGCCTGGTATCT | CTCAAGACCCGTTTAGAG | T7 |
| PPL[23] | pGEM4 | Bovine | GTGGATAACCGTATTACCGCC | CTCTGACGGCAGTTTACGAG | T7 |
| ppαF[22] | SP65 | Yeast | CCAGAAACTCAGAAGGGTCG | CAGGAAACAGCTATGAC | SP6 |
| Sec61βOPG2[22] | pcDNA5 | Human | CGCAAATGGGCGGTAGGCGTG | TAGAAGGCACAGTCGAGG | T7 |
| SMAGP-L4N | pCMV3 | Human | CGCAAATGGGCGGTAGGCGTG | TAGAAGGCACAGTCGAGG | T7 |
| SMAGP-L4N-OPG2 | pCMV3 | | | | |
| Syt1[1] | pcDNA3 | Rat | CGCAAATGGGCGGTAGGCGTG | TAGAAGGCACAGTCGAGG | T7 |
| Syt1-127-3M | pcDNA3 | | | CTACATCATCATGTCATCCTT AAGGGCCTGATCC | |
| Syt1OPG2 | pcDNA3 | | | TAGAAGGCACAGTCGAGG | |
| Syt1-127-OPG2 | pcDNA3 | | | | |
| VCAM1[1] | pcDNA3 | Human | CGCAAATGGGCGGTAGGCGTG | TAGAAGGCACAGTCGAGG | T7 |

A list of primer sequences that were used for PCR amplification to generate linear DNA templates suitable for transcription.

cells were included in translation master mixes such that each translation reaction contained $2 \times 10^5$ cells/mL.

**In vitro synthesis and membrane insertion: analysis of membrane-associated products.** Translation and membrane insertion/translocation assays (20 μL) were performed in nuclease-treated rabbit reticulocyte lysate (Promega) in the presence of EasyTag EXPRESS $^{35}$S Protein Labelling Mix containing [$^{35}$S] methionine (Perkin Elmer) (0.533 MBq; 30.15 TBq/mmol), 25 μM amino acids minus methionine (Promega), 1 μM Ipom-F, or an equivalent volume of DMSO, 6.5% (v/v) ER-derived membranes and ~10% (v/v) of in vitro transcribed mRNA encoding the relevant precursor protein[1,62]. Translation reactions using nuclease-treated canine pancreatic microsomes (from stock with $OD_{280} = 44$/mL) were performed for 20 min at 30 °C whereas those using SP HeLa cells were performed on a 1.5X scale for 1 h at 30 °C[25,28]. Following incubation with 0.1 mM puromycin for 10 min at 30 °C to ensure translation termination and ribosome release of newly synthesised proteins, microsomal or SP cell membrane-associated fractions were recovered by centrifugation through an 80 μL high-salt cushion (0.75 M sucrose, 0.5 M KOAc, 5 mM Mg(OAc)$_2$, 50 mM Hepes-KOH, pH 7.9) at $100,000 \times g$ for 10 min at 4 °C and the pellet suspended directly in SDS sample buffer (Fig. 1 and Supplementary Fig. 1).

**In vitro synthesis and membrane insertion: analysis of total translation products.** In order to define the effects of Sec61α, EMC2 or EMC5 knock-down on the in vitro ER insertion of membrane proteins, we modified our SP cell procedure to recover all translation products (i.e. membrane-associated and non-targeted nascent chains) via immunoprecipitation (Fig. 3a). This enabled us to compare the efficiency of N-glycosylation as a proportion of the total protein synthesised in each condition, thereby ruling out any effect resulting from differences in translation or minor variations in the amount of SP cells present. The addition of the OPG2-tag does not affect TMP sensitivity to Ipom-F and recovery via immunoprecipitation was consistently robust (Supplementary Fig. 3). Furthermore, inclusion of the OPG2-tag (two N-linked glycan sites) at the C-terminus of our type III TMPs (typically one N-linked glycan site) allowed us to confirm that none of these model substrates are inserted with the wrong topology ($N_{Cyt}/C_{Lum}$) under any knock-down condition, as has previously been observed with some precursors following destabilisation of the EMC[18]. On this basis, we performed translation reactions using siRNA-treated SP cells and OPG2-tagged TMP mRNA as described in the previous section. Following puromycin treatment, the total reaction material was then diluted with nine volumes of Triton immunoprecipitation buffer (10 mM Tris-HCl, 140 mM NaCl, 1 mM EDTA, 1% (v/v) Triton X-100, pH 7.5), supplemented with 5 mM PMSF and 1 mM methionine, and samples were incubated under constant agitation with appropriate antisera (1:200 dilution) for 16 h at 4 °C to recover both the membrane-associated and non-targeted nascent chains. Samples were then incubated under constant agitation with 10% (v/v) Protein-A-Sepharose beads (Genscript) for a further 2 h at 4 °C before recovery by

centrifugation at $13,000 \times g$ for 1 min. Protein-A-Sepharose beads were washed with Triton immunoprecipitation buffer prior to suspension in SDS sample buffer.

**SDS-PAGE and immunoblotting.** All samples were suspended in SDS sample buffer, treated with 1000 U of Endoglycosidase Hf or Endo H (respectively, for molecular weight proteins ~10-50 kDa or ~50–100 kDa; New England Biolabs, P0703S or P0702S) where indicated, and denatured for 1 h at 37 °C prior to resolution by SDS-PAGE (16% PAGE, 120 V, 120 min). To analyse the translation products, gels were fixed for 5 min (20% (v/v) MeOH, 10% (v/v) AcOH), dried for 2 h at 65 °C and the radiolabelled species visualised using a Typhoon FLA-700 (GE Healthcare) following exposure to a phosphorimaging plate for 24–72 h. Knock-down efficiencies (EMC2, EMC5, Sec61α, SRα) and controls (EMC6, OST, LMNB1, hSnd2, CAML) were determined by quantitative immunoblotting. Following transfer to a PVDF membrane in transfer buffer (0.06 M Tris, 0.60 M glycine, 20% MeOH) at 300 mA for 2.5 h, PVDF membranes were incubated in 1X Casein blocking buffer (10X stock from Sigma-Aldrich, B6429) made up in TBS, incubated with appropriate primary antibodies (1:500 or 1:1000 dilution) and processed for fluorescence-based detection as described by LI-COR Biosciences using appropriate secondary antibodies (IRDye 680RD Donkey anti-Goat, IRDye 680RD Donkey anti-Rabbit, IRDye 800CW Donkey anti-Guinea pig, IRDye 800CW Donkey anti-Mouse) at 1:10,000 dilution. Signals were visualised using an Odyssey CLx Imaging System (LI-COR Biosciences).

**Statistics and reproducibility.** Bar graphs depict either the efficiencies of siRNA-mediated knock-down in SP cells calculated as a proportion of the protein content when compared to the NT control or membrane insertion efficiencies calculated as the ratio of N-glycosylated protein to the amount of non-glycosylated protein, with all control samples set to 100%. Replicates, whether in vitro membrane insertion assays or immunoblot analyses of siRNA target genes, were performed using individual siRNA-mediated knock-downs ($n = 3$, biologically independent experiments). Replicates of immunoblot analyses examining the effects of siRNA-mediated knock-downs on other, non-targeted ER membrane proteins were performed using individual siRNA-mediated knock-downs ($n = 2$ or $n = 3$, biologically independent experiments) as indicated (see Supplementary Fig. 5). Normalised values were used for statistical comparison (RM one-way or two-way ANOVA; DF and F values are shown in each figure as appropriate and the multiple comparisons test used are indicated in the figure legend). Statistical significance is given as n.s., non-significant $P > 0.1$; $*P < 0.05$; $**P < 0.01$; $***P < 0.001$; $****P < 0.0001$.

**Reporting summary.** Further information on research design is available in the Nature Research Reporting Summary linked to this article.

## Data availability

The authors declare that all data supporting the findings of this study are available within the article and supplementary information files. Uncropped and unedited blot/gel images

are included in Supplementary Fig. 6. Analysis of the characteristics of Uniprot annotated type III TMPs are presented in Supplementary Data 1. All numerical data and statistical analyses are available in Supplementary Data 2.

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

## Acknowledgements

For gifting reagents, we thank Cornelia Wilson (CD3δ cDNA; Canterbury Christ Church University, Kent, UK), Martin Spiess (ASGR1 cDNA; University of Basel, Basel, Switzerland), Jeffrey Brodsky (ppαF cDNA; University of Pittsburgh, Pittsburgh, PA), Sven Lang and Richard Zimmermann (Sec61α and hSnd2 antisera; University of Saarland, Homburg, Germany) and Martin Pool (SRα antisera; University of Manchester). We additionally thank Pawel Leznicki, Martin Pool and Lisa Swanton (University of Manchester, Manchester, UK) and Joen Luirink (Free University, Amsterdam, Netherlands) for critical feedback during manuscript preparation and revision. This work was supported by a Welcome Trust Investigator Award in Science 204957/Z/16/Z (S.H.) and an AREA grant 2R15GM116032-02A1 from the National Institute of General Medical Sciences of the National Institutes of Health (NIH) and a Ball State University (BSU) Provost Startup Award (W.Q.S.). L.E.A. is the recipient of BSU honors college research grants from 01/2020 to 12/2020.

## Author contributions

G.Z., K.B.D. and L.E.A performed the chemical synthesis of Ipom-F and W.Q.S supervised the synthesis. S.O'K. performed all experiments under the supervision of S.H. S. O'K. and S.H. designed the experiments, analysed the data and wrote the manuscript with the approval from all authors.

## Competing interests

The authors declare no competing interests.
