## [Transparent Peer Review File · Communications Biology]

Reviewers' comments:

Reviewer #1 (Remarks to the Author):

Sec61 complex is a major site for membrane insertions of newly synthesized proteins. However, authors group and others previously showed that Ipomoeassin F (Ipom-F) and mycolactone, the inhibitors for Sec61-mediated translocation, did not affect two type III TMDs (GypC and Syt1). Recently, ER membrane complex (EMC) was shown to act as insertase. Thus, in this manuscript, the authors investigated the requirement of EMC (also Sec61 and SRP) for membrane integration of type III TMDs.

Authors used sophisticated biochemical methods to investigate Sec61 and EMC dependence for membrane integrations of five type III TMDs. They first used a Sec61-dependent Type I transmembrane protein and an EMC-dependent Tail-anchor protein as a positive and negative controls and have proven these methods works well and results are reliable. Authors showed four of five type III TMDs showed clear EMC-dependence. Moreover, Authors showed that the integration defects of three type III TMDs in EMC depletion were strongly enhanced by additional Sec61 depletion. Thus, the biogenesis of type III TMDs involves both the EMC and the Sec61 complex. Authors proposed that the Sec61 complex facilitates the membrane insertase activity of the EMC, rather than directory act on the membrane insertion of Type III TMDs, because of consistent insensitivity of Ipom-F. Finally, they showed the requirement of SRP for membrane integrations of five type III TMDs.

Overall, this paper's results are convincing, and their findings are interesting and many researchers in the field of membrane proteins biogenesis would show the interest on these findings. Only shortage of the manuscript is the part of SRP: I feel the part of SRP is not connected the whole story well and isolated. To understand the relationship of SRP and Sec61/EMC more clearly, I hope to see the effects of the double or triple deficiencies of SRP and Sec61/EMC on the membrane insertion of Type III TMDs. Are the effects additive, synergic or epistatic? With these results, please add a schematic to show the relationship between SRP and Sec61/EMC to the model in Figure 9.

Reviewer #2 (Remarks to the Author):

The manuscript by O'Keefe et al focuses on understanding the pathway by which type III single-pass membrane proteins are integrated in the ER membrane. Most membrane proteins and inserted via the canonical Sec61 complex, which acts co-translationally on proteins delivered to the ER by the signal recognition particle (SRP). Some membrane proteins, most notably so-called "tail-anchored" membrane proteins utilize alternative pathways, including the GET pathway and EMC. More recently, the EMC has been shown to coordinate with the Sec61 complex during biogenesis of certain multi-pass membrane proteins whose first signal-anchor TMD adopts an Nexo orientation (Chitwood & Hegde 2018). Accordingly, the EMC facilitates the headfirst (Nexo) integration of the first TMD into the bilayer, while downstream TMDs are integrated by Sec61. Consistent with the idea that Sec61 cannot efficiently integrate Nexo signal-anchor TMDs, small molecule Sec61 inhibitors including ipomoeassin-F potently inhibit the integration of type I and type II single-pass proteins, but not type III single-pass proteins that adopt the Nexo/Ccyt orientation in the membrane. Here the authors explore the role of Sec61 and EMC in the biogenesis of these type III single-pass membrane proteins.

The authors first use in vitro systems (canine microsomes and semi-permeabilized HeLa cells,) to demonstrate that five different type III proteins (and one TA protein) are, as expected, resistant to Ipom-F treatment, whereas integration of type I and type II proteins is blocked. Next, they find that while siRNA depletion of Sec61 impaired integration of the type II protein ASGR1, it has no effect on integration of four of five type III proteins. Conversely, EMC depletion selectively impaired four of the same five type III proteins (and one TA protein), but had no effect on the type II protein ASGR1. From

this they conclude that many type III proteins do indeed require EMC for efficient integration. Next, the authors show that double Sec61 and EMC knock-downs show further reduction in type III membrane insertion, suggesting that Sec61 and EMC may coordinate during type III membrane protein insertion. Finally, the authors confirm a role for SRP in targeting type III membrane proteins to the EMC.

Overall this is a well-executed and technically sound study. However, most of what is shown here confirms previous work (Chitwood et al., 2018), which established a role for EMC (and SRP/SR) in Nexo signal anchor integration in certain multi-pass and artificial type III substrates. The present study modestly extends this work by formally demonstrating EMC-dependence of bona fide type III proteins. The observation that double knockdowns of Sec61 and EMC show stronger phenotypes than the individual depletions for a number of type III proteins is also consistent with this previous work. However, unlike in that system (which focused on multi-pass GPCRs with Nexo TM1), single-pass type III proteins have only a single TMD, making it harder to rationalize the need for both Sec61 and EMC. Perhaps this is related to the observation that the length of the cytosolic domain of Syt1 influences its EMC-dependence, but the authors don't follow up on this initial observation.

Additional comments:

- The manuscript is quite long and could easily be shortened.
- The abstract mentions a "non-canonical Sec61 translocon" but it is not clear exactly what this means. This should be explicitly defined, or removed.
- Fig. 9—the recent cryoEM structures of EMC suggest that it is unlikely to closely approach a Sec61-engaged ribosome because there would not be enough space to accommodate its large cytosolic domain between the ribosome and the membrane. This should be made obvious in the cartoon
- The abbreviations "TMP" and "TMD" are so similar as to be confusing at times.
- Sup. Fig. 6A—The topology cartoons for WRB and CAML are incorrect. WRB is Nexo/Ccyt, while CAML has three TMDs and is Ncyt/Cexo. Also, why is the third TMD of WRB shown as a "tail-anchor"?

Professor Stephen High
School of Biological
Sciences
The University of
Manchester
Oxford Road
Manchester. M13 9PT.

+44(0)161 275 5070
www.manchester.ac.uk

20th April 2021

We thank our two Reviewers for their thoughtful and constructive comments regarding our original submission and provide a detailed response to each of the points that they raised below.

Reviewer #1 (Remarks to the Author):

Sec61 complex is a major site for membrane insertions of newly synthesized proteins. However, authors group and others previously showed that Ipomoeassin F (Ipom-F) and mycolactone, the inhibitors for Sec61-mediated translocation, did not affect two type III TMDs (GypC and Syt1). Recently, ER membrane complex (EMC) was shown to act as insertase. Thus, in this manuscript, the authors investigated the requirement of EMC (also Sec61 and SRP) for membrane integration of type III TMDs.

Authors used sophisticated biochemical methods to investigate Sec61 and EMC dependence for membrane integrations of five type III TMDs. They first used a Sec61-dependent Type I transmembrane protein and an EMC-dependent Tail-anchor protein as a positive and negative controls and have proven these methods works well and results are reliable. Authors showed four of five type III TMDs showed clear EMC-dependence. Moreover, Authors showed that the integration defects of three type III TMDs in EMC depletion were strongly enhanced by additional Sec61 depletion. Thus, the biogenesis of type III TMDs involves both the EMC and the Sec61 complex. Authors proposed that the Sec61 complex facilitates the membrane insertase activity of the EMC, rather than directory act on the membrane insertion of Type III TMDs, because of consistent insensitivity of Ipom-F. Finally, they showed the requirement of SRP for membrane integrations of five type III TMDs.

Overall, this paper's results are convincing, and their findings are interesting and many researchers in the field of membrane proteins biogenesis would show the interest on these findings. Only shortage of the manuscript is the part of SR α : I feel the part of SR α is not connected the whole story well and isolated.

Comments	Response
1) To understand the relationship of SRP and Sec61/EMC more clearly, I hope to see the effects of the double or triple deficiencies of SRP and Sec61/EMC on the membrane insertion of Type III TMDs. Are the effects additive, synergic or epistatic?	We have now investigated the effects of double knockdowns of SR α +EMC5 and SR α +Sec61 α on control and type III TMPs (see revised Fig. 6). In each case, the effect of these combined knockdowns leads to an equivalent or enhanced insertional defect as compared to a knockdown of EMC5

	alone. For 5 of the 6 type IIIs tested (ie. those with a relatively short cytosolic domain length), the insertional defect of SRα+EMC5 knockdown mirrors that seen following knockdown of Sec61α+EMC5. Given these data (Fig. 6) and the effects of these new combined knockdowns on other ER protein translocation components (see revised Supplementary Fig. 5) we conclude that the effects are additive, and not synergistic or epistatic. We also attempted a triple knockdown of all three components, but this resulted in extremely high levels of cell death and we did not to pursue this approach any further.
2) With these results, please add a schematic to show the relationship between SRP and Sec61/EMC to the model in Figure 9.	We have updated our model to reflect these new data. Please see revised Fig. 7.

Reviewer #2 (Remarks to the Author):

The manuscript by O’Keefe et al focuses on understanding the pathway by which type III single-pass membrane proteins are integrated in the ER membrane. Most membrane proteins are inserted via the canonical Sec61 complex, which acts co-translationally on proteins delivered to the ER by the signal recognition particle (SRP). Some membrane proteins, most notably so-called “tail-anchored” membrane proteins utilize alternative pathways, including the GET pathway and EMC. More recently, the EMC has been shown to coordinate with the Sec61 complex during biogenesis of certain multi-pass membrane proteins whose first signal-anchor TMD adopts an Nexo orientation (Chitwood & Hegde 2018). Accordingly, the EMC facilitates the headfirst (Nexo) integration of the first TMD into the bilayer, while downstream TMDs are integrated by Sec61. Consistent with the idea that Sec61 cannot efficiently integrate Nexo signal-anchor TMDs, small molecule Sec61 inhibitors including ipomoeassin-F potently inhibit the integration of type I and type II single-pass proteins, but not type III single-pass proteins that adopt the Nexo/Ccyt orientation in the membrane. Here the authors explore the role of Sec61 and EMC in the biogenesis of these type III single-pass membrane proteins.

The authors first use in vitro systems (canine microsomes and semi-permeabilized HeLa cells,) to demonstrate that five different type III proteins (and one TA protein) are, as expected, resistant to Ipom-F treatment, whereas integration of type I and type II proteins is blocked. Next, they find that while siRNA depletion of Sec61 impaired integration of the type II protein ASGR1, it has no effect on integration of four of five type III proteins. Conversely, EMC depletion selectively impaired four of the same five type III proteins (and one TA protein), but had no effect on the type II protein ASGR1. From this they conclude that many type III proteins do indeed require EMC for efficient integration. Next, the authors show that double Sec61 and EMC knock-downs show further reduction in type III membrane insertion, suggesting that Sec61 and EMC may coordinate during type III membrane protein insertion. Finally, the authors confirm a role for SRP in targeting type III membrane proteins to the EMC.

Overall this is a well-executed and technically sound study.

Comments	Response
1) However, most of what is shown here confirms previous work (Chitwood et al., 2018), which established a role for EMC (and SRP/SR) in Nexo signal anchor integration in certain multi-pass and artificial type III substrates. The present study modestly extends this work by formally demonstrating EMC-dependence of bona fide type III proteins. The observation that double knockdowns of Sec61 and EMC show stronger phenotypes than the individual depletions for a number of type III proteins is also consistent with this previous work.	We agree that our study confirms previous work that established a role for SRP/SR and the EMC in the insertion of artificial type III substrates and Nexo signal anchor integration for certain GPCRs and (Chitwood et al., 2018). However, in contrast to this previous work, our study uses a panel of bona fide type III TMPs to dissect the mechanism by which Nexo signal anchors are inserted via the EMC. Hence, we show that following SRP-dependent delivery to the ER membrane, the integration of single spanning type III TMPs involves the insertase activity of the EMC acting in concert with a non-insertase contribution from the Sec61 complex. The concerted actions of Sec61/EMC in the insertional route that we propose here is likely also responsible for the integration of the Nexo signal anchor of certain GPCRs, whilst their downstream transmembrane domains utilise the insertion activity of the Sec61 complex. In short, our work provides new and significant insights into the biogenesis of type III TMPs that has been further enhanced by the additional experiments we have performed in response to the comments to Reviewer 1.
2) However, unlike in that system (which focused on multi-pass GPCRs with Nexo TM1), single-pass type III proteins have only a single TMD, making it harder to rationalize the need for both Sec61 and EMC. Perhaps this is related to the observation that the length of the cytosolic domain of Syt1 influences its EMC-dependence, but the authors don't follow up on this initial observation.	Our study clearly shows that for single-spanning type III TMPs, the characteristics that typically confer EMC-dependence to tail-anchor and certain multi-pass TMPs (moderate-low TMD hydrophobicity and/or charged/polar residues) are not applicable. Rather, we propose that it is the need for type III TMPs to co-ordinate the translocation of a 'pre-made' N-terminus with the integration of their transmembrane domain that most likely necessitates the need for both Sec61 and the EMC (see lines 377-389 of the revised manuscript highlighted in yellow). How the length of the cytosolic domain influences the EMC-dependence of type III

	TMPs remains an open question that we plan to address further in a future study (see lines 399-406 of the revised manuscript).
3) The manuscript is quite long and could easily be shortened.	In addition to including new data, the manuscript word count has been shortened by ~8% (~800 words under the maximum article length) and the number of main Figures reduced from 9 to 7.
4) The abstract mentions a “non-canonical Sec61 translocon” but it is not clear exactly what this means. This should be explicitly defined, or removed.	The term “Non-canonical” has been removed from the abstract and the relevant text amended. See lines 24-28 of the revised manuscript.
5) Fig. 9—the recent cryoEM structures of EMC suggest that it is unlikely to closely approach a Sec61-engaged ribosome because there would not be enough space to accommodate its large cytosolic domain between the ribosome and the membrane. This should be made obvious in the cartoon	The relative size of the ribosome in our schematic has been increased and the proposed minimum distance between the Sec61 lateral gate and the insertase site of the EMC has been annotated. See revised figure 7b, route 2, pink arrow.
6) The abbreviations “TMP” and “TMD” are so similar as to be confusing at times.	To reduce confusion, we have now removed the ‘TMD’ abbreviation throughout the manuscript, using ‘transmembrane domain’ instead. All instances are highlighted in green.
7) Sup. Fig. 6A—The topology cartoons for WRB and CAML are incorrect. WRB is Nexo/Ccyt, while CAML has three TMDs and is Ncyt/Cexo. Also, why is the third TMD of WRB shown as a “tail-anchor”?	Thank you for flagging these errors which have now been corrected. The topology cartoon for hSnd2 has also been updated based on information within Lei et al. 2020 (DOI: 10.1093/abbs/gmz157) and topology prediction software. See revised Supplementary Fig. 5.

Updated Figures

Changes made to Manuscript figures:

- Original Fig. 2 moved into supplementary – see new Supplementary F1.
- Fig. 2, inclusion of dot-plot to show data distribution.
- Fig. 3, inclusion of dot-plot to show data distribution.
- Fig. 4, inclusion of dot-plot to show data distribution.
- Original Fig. 6 has been integrated into the previous figure – see new Fig. 5.
- Fig. 6, inclusion of new data investigating the effects of double knockdowns of SR α +EMC and SR α +Sec61 α on control and type III TMPs.
- Fig. 7, inclusion of a schematic showing the relationship between SRP and Sec61/EMC to our model.

Supplementary Data:

- Original Fig. 2 has been integrated into supplementary – see new Supplementary F1.
- Original Supplementary Fig. 4 has been integrated into the previous figure – see new Supplementary Fig. 3.
- Original Supplementary Fig. 7 has been integrated into the previous figure – see new Supplementary Fig. 5.

Fig. 2

To comply with formatting requirements, the bar graph (part d) now includes a dot-plot to show data distribution.

Fig. 3

To comply with formatting requirements, the bar graph (part d) now includes a dot-plot to show data distribution.

Fig. 4

To comply with formatting requirements, the bar graph (part d) now includes a dot-plot to show data distribution.

Fig. 5

To shorten the manuscript, comparison of the effect of double knock-downs to single knock-downs for ASGR1 and Cytb5 has been removed, the effect on type III TMPs (former Fig. 7) has been integrated into this figure (parts e-f) and the table from former Fig. 7 removed. To comply with formatting requirements, bar graphs (parts b, d, f) now include dot-plots to show data distribution.

Fig. 6

To understand the relationship of SRP and Sec61/EMC more clearly, new data showing the effects of double knockdowns of SR α +EMC and SR α +Sec61 α on control and type III TMPs has now been incorporated into this Figure (parts b-d). These effects are compared with the insertional defects observed in previous knockdowns (part e) to allow readers to clearly see that the effects of these double depletions are additive.

Fig. 7 A schematic showing the relationship between the SRP/SRP receptor complex and the Sec61/EMC ER insertion site has been added to our model. The ribosome size has been increased and the suggested minimum distance between the Sec61 lateral gate and the insertase site of the EMC is now indicated on route 2.

Supplementary Fig. 1

To shorten the manuscript, the original Fig. 2 of the main text is now part E of this Figure. The analysis of li insertion into microsomes shown in the in the original version of this Figure (panel Di) has been removed since it is duplicated in panel E of the revised Figure above.

Supplementary Fig. 3

To help shorten the manuscript, the original supplementary Fig. 4 is now included as part C of this Figure. To comply with formatting requirements, the bar graph (part C) now includes a dot-plot to show data distribution.

Supplementary Fig. 5

To understand the relationship of SRP and Sec61/EMC more clearly, the effects of double knockdowns of SR α +EMC and SR α +Sec61 α on ER protein translocation components have been included. To shorten the manuscript, the effects of knockdowns on SRP54 (original Supplementary Fig. 7) have now been merged incorporated here. To comply with formatting requirements, bar graphs (part C) now include a dot-plots to show data distribution.

REVIEWERS' COMMENTS:

Reviewer #1 (Remarks to the Author):

A satisfactory revision was done. I suggest to accept for publication.

Reviewer #2 (Remarks to the Author):

I am satisfied with the author's response to the original reviews, and support publication of their manuscript.